# Quality of life in the postpartum period of Mexican women living with HIV: The role of clinical and sociodemographic factors

Mauricio Domínguez-Castro[1][☯], Alicia Ramírez-Ramírez[2][☯],
Noemí Guadalupe Plazola-Camacho[2], Miroslava Avila-García[2‡], Edna Basilio-Gálvez[2‡],
Margarita López-Martínez[1], Lucky Norah Katende-Kyenda[3], Ismael Mancilla-Herrera[4],
Diana Mercedes Soriano-Becerril[2], José Romo-Yáñez[5], Carmen Selene García-Romero[2],
Ricardo Figueroa-Damián[2], Jessica Hernández-Pineda[2]*

1 Department of Physiology and Cell Development, National Institute of Perinatology, Miguel Hidalgo, Mexico City, Mexico, 2 Department of Infectology and Immunology, National Institute of Perinatology, Miguel Hidalgo, Mexico City, Mexico, 3 Department of Internal Medicine and Pharmacology, Walter Sisulu University, School of Medicine, Faculty of Health Sciences, Mthatha, Eastern Cape, South Africa, 4 Deputy Directorate of Biomedical Research, National Institute of Perinatology. Miguel Hidalgo, Mexico City, Mexico, 5 Coordination of Gynecological and Perinatal Endocrinology, National Institute of Perinatology. Miguel Hidalgo, Mexico City, Mexico

☯ These authors contributed equally to this work.
‡ These authors also contributed equally to this work.
* jesspinq@yahoo.com.mx

## Abstract

### Introduction

The HIV epidemic remains a major public health challenge, specifically for women living with HIV who face vulnerabilities during pregnancy and motherhood. Furthermore, high vulnerability is closely linked to a lower quality of life. It is essential to address the determinants that influence quality of life within this population in order to enhance health outcomes and inform the development of evidence-based care protocols.

### Methods

An observational, cross-sectional study was conducted from 2020 to 2022 to assess the quality of life of Mexican postpartum women living with HIV (n = 75). The WHOQoL-HIV-Bref instrument was used. Quality of life results were analysed statistically in relation to sociodemographic and clinical factors to evaluate their associations and their predictive power through multinomial logistic regression analysis.

### Results

Nearly half of postpartum WLWH experienced a lower quality of life (49%). Psychological health, Environment, Spirituality, religion and personal beliefs domains scores

**Data availability statement:** The database, which includes sociodemographic and pathophysiological variables from a cohort of 75 postpartum women (collected between 2020–2022), has been deposited in the Zenodo repository to ensure open access and long-term preservation. Please find the repository information below: Database Title: Availability total dataset _QoL in the Mexican postpartum WLWH Repository: Zenodo DOI: 10.5281/zenodo.19895187 Access Link: https://zenodo.org/records/19895187 Domínguez-Castro, M., Ramírez-Ramírez, A., & Hernández-Pineda, J. (2026). Availability total dataset _QoL in the Mexican postpartum WLWH [Data set]. In PLoS ONE. Zenodo. https://doi.org/10.5281/zenodo.19895187.

**Funding:** This study was supported by the Instituto Nacional de Perinatología (Mexico City) under approval number 212250-3120771. The founders had no role in study design, data collection and analysis, decision to publish, or preparation of the manuscript.

**Competing interests:** The authors have declared that no competing interests exist.

were below the median. There were multiple associations with socio-demographic and clinical factors. Highlights the use of ART during pregnancy and postpartum, HIV symptoms, sexual behavior, marital and educational status, mainly. Physical health, Independence level, and Spirituality, religion and personal beliefs domains were identified as potential predictors of a perception of high quality of life of this population.

## Conclusion

There are clinical and socio-demographic factors that influence the perception of quality of life in Mexican women living with HIV during postpartum. It is important to identify and mitigate these factors for the well-being of these women and their children. Further research is needed to evaluate the impact of interventions on Physical health, Independence level, and Spirituality, religion and personal beliefs domains to improve perception of high quality of life in WLWH. These findings ultimately underscore the need to integrate quality of life assessment as the fourth '95' target in World Health Organization strategies for people living with HIV.

## Introduction

The HIV epidemic continues to be one of the most significant public health challenges after more than 40 years of its emergence. To date, people living with HIV (PLWH) face challenges related to HIV infection, even when they have had sustained antiretroviral therapy (ART) and they achieve suppression of viral load (VL) of HIV [1–3]; two of the three 95-95-95 important UNIADS targets for eradicating HIV pandemic [4–6]. From a clinical perspective, PLWH are confronted with the chronic nature of the infection, the long-term effects of ART and the associated challenges of adherence. They also experience the burden of non-infectious comorbidities [7]. In addition to these clinical demands, PLWH also face persistent stigma and discrimination. This should be understood as an additional challenge that compounds the complexity of living with HIV, rather than being a direct clinical manifestation [2]. Together, these clinical and social factors have a significant impact on perception of health and their QoL [8,9]. This impact is particularly pronounced in highly vulnerable groups, such as WLWH, who may encounter additional stressors during pregnancy, the postpartum period, and motherhood [10]. During these periods, pain and physical discomfort, combined with environmental factors, social relationships and personal beliefs, can exacerbate vulnerability and perpetuate stigma and discrimination. Consequently, perceptions of health status deteriorate, QoL declines and negative outcomes such as reduced self-care and diminished attention may ensue [11,12].

There are also significant associations between maternal postpartum health (both mental and physical) and the health of their children [13]. Therefore, a mother's QoL could influence her children's QoL as they grow up. It is known that even in non-complicated perinatal period important physical and emotional changes occurs, and factors such as age, parity, social and economic issues, obstetric complications,

obesity, history of alcohol dependence, sleep difficulties, stress, anxiety and postnatal depression affect the mother QoL [13–15]. Furthermore, studies assessing QoL in mothers with chronic diseases have reported that their children also experience lowest QoL [13,16]. Another study evaluating health-related quality of life (HR-QoL) in mother-child pairs from low-income settings found that these children had poorer outcomes and behavioural disorders compared to children in the general population [17]. It is important to emphasise that the traditional approach to health has primarily focused on the clinical dimension, centered on the detection, treatment and cure of diseases. However, health cannot be reduced to the mere absence of illness as it permeates every aspect of a patient's life, making this approach insufficient. Therefore, the assessment of QoL emerges as a valuable indicator of self-perceived well-being and health status, encompassing broader dimensions of QoL [18].

QoL has several definitions and instruments to assess it [18]. Furthermore, according to experts, the assessment of QoL should be evaluated with specialised and validated instruments for a population, especially if it involves a population with chronic diseases [19,20]. According to the World Health Organisation (WHO), QoL is the perception that a person has of their position in life, in the context of the culture and value system in which they live, and concerning their objectives, expectations, standards and concerns, and provides a QoL instrument for the PLWH [8].

Expert groups on HIV/AIDS have proposed to include the QoL evaluation in PLWH as a measure of therapeutic success, as well as a 'fourth 95' HIV target which should be achieved [8]. QoL is increasingly recognised as a key component of public health [21], guiding management strategies and informing the development of new health policies for PLWH. This approach is particularly important for vulnerable groups, such as WLWH, as it helps to improve medical care and address their specific needs [7,9,10].

Globally, WLWH represent 53% of all PLWH [22], and the majority are of reproductive age. This increases the likelihood of pregnancy and the risk of exposing their newborns to HIV vertical transmission [23]. Around 1.2 million WLWH become pregnant each year, and 84% of these (72–98%) receive ART to prevent mother-to-child-transmission (MTCT) [23]. Therefore, aligning with important United Nations Sustainable Development Goals, particularly the Goal 3: Good Health and Well-being, within the context of countries facing significant challenges related to social inequalities, is necessary.

Significant progress has been made in understanding the QoL of pregnant and postpartum WLWH in Latin America (LATAM), particularly in Brazil [24–26]. Nevertheless, important gaps remain that could be addressed by future research, particularly by moving beyond a purely medical perspective. There is a need for greater exploration of factors such as emotional impact, social stigma and support networks, as well as the population characteristics that shape these experiences. This is essential to better understand how these women navigate their condition and the demands of motherhood.

The objectives of this study were to describe the perception of QoL during postpartum period of Mexican WLWH and to determine its association sociodemographic and physiopathological factors.

## Materials and methods

### Study design

An observational, cross-sectional study to evaluate the perception of QoL of postpartum WLWH was conducted at the Instituto Nacional de Perinatología (INPer) in Mexico City, a renowned Institution for pregnancy and neonatal care in WLWH. The study was reviewed and registered by the Ethics and Research Committees of the Institute with number 212250−3120771 and was carried out for recruiting patients between December 09th, 2020 and June 17th, 2022. The sample size for assessing QoL was calculated to include 28 women using a probability finite sample for qualitative variables ($Z^2_a = 1.96$, p = 0.5, $d^2 = 5\%$). We considered 50% losses in recruitment during the study due to irregularities in attention during COVID-19 pandemic. The final sample size calculated was n = 42. With the objective to evaluate sociodemographic and physio pathological factors that influence the QoL, a second sample size was calculated considering n = 75 women by multinomial linear regression (MLR) analysis, considering 15 significant variables by Spearman's correlation analysis with

acceptable power, using Cohen's f² as the effect size measure. After delivery, postpartum WLWH were referred to their urban medical center. It just came back only for their baby's follow-up until obtaining discharge, **upon confirmation of no** HIV **vertical transmission** (18 months).

### Women enrolment

Sampling was conducted using a non-probabilistic convenience approach. Postpartum WLWH receiving ART treatment as a part of their infection care, over 18 years old, with prenatal care at the INPer, were invited to participate during the visits of baby´s follow-up. Those who did not agree to participate received their babies medical care as usual. The participants who accepted signed a written informed consent; their anonymity and privacy were protected. Participants returned to our facilities for a clinical history that included sociodemographic, nonpathological diseases, other infectious diseases, obstetric history, sexual behaviours. The QoL instrument was administered on a single occasion to each postpartum WLWH. The data were collected at varying postpartum times and among a diverse sample of postpartum WLWH.

### Quality of life assessment

QoL was assessed using the Spanish version of WHOQoL-HIV-Bref [27], a short version of the WHOQoL-HIV instrument designed for PLWH, which has been validated in Spain and Latin America (LATAM) [24,28]. The HIV-specific items extracted from WHOQoL-HIV long form were translated and integrated into the Spanish version of WHOQoL-Bref to complete the 31 items of the WHOQoL-HIV-Bref instrument [27], ensuring the instrument's relevance and reliability in our study.

This instrument measures the QoL through a 31-item scale grouped into six domains as follows: 4-items on Physical health, 5-item on Psychological health, 4-items on independence level, 4-items on Social relationships, 8-items on Environment, 4-items on Spirituality, religion and personal beliefs and 2-items measure the total or general QoL through the rate of perception of QoL and satisfaction with their health [18]. Individual items are rated on a Likert scale where 1 indicates low/negative perceptions, and 5 indicates high/positive perceptions. Items that ask about negative perceptions and experiences are reverse-coded for scoring. Therefore, higher scores for all items indicated better QoL. The average score for each domain was multiplied by four, producing domain scores ranging from 4 to 20, where 4 means the lowest QoL and 20 is the highest [27].

In light of previous studies, the median of total QoL score was utilised as a reference point to categorise the QoL perceived by the participants in this study. Subsequently, the QoL was categorised as low, medium, or high, according with this reference point [29,30].

Sociodemographic information was added to the instrument and consisted of 11 questions: type of employment, monthly income, and residence in a rural or urban zone. Additionally, information about clinical data on comorbidities, CD4 count, viral load, type of ART, dosage, and adherence was also included. Eight specific questions related to gestational age, postpartum time, perception of complications in these periods, and support from social networks were added. Our instrument was validated by five HIV specialist partners, four Spanish speakers, and one English-Spanish speaker (S1 File). Patients identified as having physical or psychological problems during the intervention were referred to the specialty for additional medical care.

### Statistical analysis

Means and standard deviations for continuous variables were reported. For categorical variables, the data are presented as frequencies and percentages. The Jarque-Bera test was performed to determine the normality of the data obtained, and then nonparametric analysis was applied. Spearman's correlation coefficient was used to determine the association of QoL instrument scores with demographic, nonpathological history, infectious history and ART management, obstetric history, and sexual behavior variables.

U-Mann–Whitney and Kruskal–Wallis tests were performed, Dunn's and Bonferroni's post-hoc analysis tests were used. A statistical significance of $p \leq 0.05$ was considered in all cases. The Cronbach index (α) was calculated to determine the internal consistency or reliability of the WHOQoL-HIV-Bref instrument. From the α-index range (0–1), the acceptable benchmark value considered was 0.7, which is acceptable from 0.7 to 0.79. Statistical analyses were performed using GraphPad Prism v7 (Dotmatics, UK).

To identify variables most closely related to QoL and evaluate their performance as potential predictors, a multinomial logistic regression (MLR) analysis was conducted on the complete dataset. This analysis adhered to all model assumptions, using "high quality of life" as the reference category. Variables exhibiting an odds ratio (OR) > 2 were selected for further consideration. Additionally, a Principal Component Analysis (PCA) was performed on the entire dataset to identify variables accounting for 60% of the total variance. The results of the MLR and PCA were cross-referenced; variables with an OR > 2 and an absolute loading value ≥ 0.6 were then tested for multicollinearity by a linear regression, with a Variance Inflation Factor (VIF) < 5 and a Tolerance < 0.7 required for inclusion. To determine potential predictors of QoL, selected variables were evaluated using Area Under the Receiver Operating Characteristic (AUC-ROC) curve analysis for the low, medium, and high categories independently. Variables with an AUC ≥ 0.7 and a $p \leq 0.05$ were chosen for the calculation of specificity, sensitivity, and the optimal cutoff value via the Youden Index. Confidence intervals (95% CI) were determined using the Wilson Score Interval. Finally, a separate PCA was performed for each QoL category to identify the specific variables accounting for 60% of the variance within each group. A statistical significance of $P \leq 0.05$ was considered in all cases. Statistical analyses were performed using GraphPad Prism v7 (Dotmatics, UK), and SPSS Statistics 26 (IBM, NY, USA).

## Results

### Characteristics of the Mexican postpartum WLWH population in the study

The sample consisted of 75 postpartum WLWH with an average age of 29 ± 5.4 years. They reported several ART regimens as a part of their HIV clinical management during pregnancy and postpartum. Upon admission to prenatal care and delivery at INPer, most of them were in the third trimester of pregnancy, averaging 32 ± 8 gestational weeks. Among these women, 76% had completed nine years of education, 47% cohabited with their partners, 69% were housewives, and 92% resided in urban areas, as presented in Table 1. Monthly incomes ranged from USD $28.4 to USD $1705.55, placing 97% in the low-income classification according to the National Institute of Statistics and Geography (INEGI, Mexico) [31].

Regarding drug use during their current pregnancies, participants denied current consumption of alcohol, tobacco, or illegal substances. However, they reported alcohol consumption in 21%, illegal drug use in 8%, and minimal cases of combined use (6%) histories, as detailed in Table 1.

Concerning the partner's drug addiction, 17% of woman denied knowledge of their current partner's substance use. The remaining 83% of the woman reported smoking (3%), alcoholism (12%), illegal drug addiction (7%), combined alcohol and illegal drug use (14%), and a mix of alcohol, smoking, and illegal drugs (5%) among those aware of it. Regarding tattoos and piercings, 25% had tattoos, 5% had piercings, and 9% reported having both (Table 1).

In terms of sexual behavior, most of the participants initiated sexual activity in late adolescence, with a history of 3–5 sexual partners until the study period. Unfortunately, 9% reported having been sexually abused. They were multigravida, delivered by Caesarean section, and reported pregnancy loss (0.55 ± 1.04) based on their gynaecological and obstetric history (Table 1).

The percentage of our participants diagnosed with HIV during their current pregnancy was 32%, and most of these cases believed that it was sexually transmitted (Table 2).

All participants received ART immediately after the diagnosis and throughout their pregnancies with effective clinical management, as indicated by CD4 + levels and undetectable viral loads, preventing vertical transmission (Table 2). The

**Table 1. Demographic characteristics, obstetric history and sexual behavior of the Mexican post-partum WLWH.**

| Characteristic | Mean±*SD* [range] Frequency (%) n=75 |
|---|---|
| **Age (years)** | 29±5.4 [19–41 years] |
| **Gestational age at medical admission (weeks)** | 32±8 [9-40.2] |
| First Trimester (1–12 gweeks) | 2 (3) |
| Second Trimester (13–26 gweeks) | 18 (24) |
| Third Trimester (27–40 gweeks) | 55 (73) |
| **Educational status (years)** | |
| ≤ 6 years | 14 (19) |
| 9 years | 36 (48) |
| 12 years | 21 (28) |
| >12 years | 4 (5) |
| **Marital status** | |
| Single | 22 (29) |
| Married | 11 (14) |
| Cohabiting | 35 (47) |
| Separated | 5 (7) |
| Widow | 2 (2) |
| **Employment type** | |
| Permanent | 11 (15) |
| Casual | 12 (16) |
| Housewife | 52 (69) |
| **Place of residence** | |
| Urban | 69 (92) |
| Rural | 6 (8) |
| **Place where they come from** | |
| Mexico City | 27 (36) |
| Metropolitan area | 45 (60) |
| Other state | 3 (4) |
| | n=73 |
| **Monthly income (USD$)** | USD$258.08±223.76 [28.41−1,705.55] |
| None[a] | 3 (4) |
| <58–88 | 7 (10) |
| 89–175 | 18 (25) |
| 176–351 | 34 (47) |
| 352–585 | 8 (11) |
| >586 | 3 (4) |
| **Year of interview** | |
| 2020 | 20 (27) |
| 2021 | 49 (65) |
| 2022 | 6 (8) |
| **Smoking habit** | n=74 |
| Yes | 22 (30) |
| No | 52 (70) |
| **Alcoholism and drugs addiction** | n=74 |
| None | 49 (66) |

*(Continued)*

**Table 1.** (Continued)

| Characteristic | Mean±*SD* [range] Frequency (%) n=75 |
|---|---|
| Alcoholism (in any degree) | 15 (20) |
| Illegal drugs addiction | 6 (8) |
| Both | 4 (5) |
| **Tattoos and piercings** | n=57 |
| None | 35 (61) |
| Tattoos | 14 (25) |
| Piercings | 3 (5) |
| Both | 5 (9) |
| **Partner's addictions** | n=58 |
| None | 23 (40) |
| Smoking | 2 (3) |
| Alcoholism (in any degree) | 7 (12) |
| Illegal drugs addiction | 4 (7) |
| Alcoholism and drugs addiction | 8 (14) |
| Alcoholism and smoking | 1 (2) |
| All above | 3 (5) |
| Do not know | 10 (17) |
|  | n=72 |
| **Age of beginning of sexual life (years)** | 17±2 [11 −25] |
| <15 | 9 (13) |
| 15–17 | 40 (56) |
| ≥18 | 22 (31) |
|  | n=73 |
| **Number of sexual partners** | 4±4 [1-25] |
| 1–2 | 22 (30) |
| 3–5 | 40 (55) |
| 6–10 | 7 (10) |
| >10 | 4 (5) |
| **Obstetric history** |  |
| Gestations | 2.61±1.46 [1–7] |
| Labour | 0.91±1.02 [0-3] |
| Caesarean section | 0.57±0.8 [0-4] |
| Abortions | 0.55±1.04 [0-5] |
| **Time of postpartum (months)** |  |
| Less than 12 months | 55 (73) |
| More than 12 months | 20 (27) |
| **Current Birth Delivery** |  |
| Labour | 10 (13) |
| Caesarean section | 65 (87) |
| **Sexual abuse history** | n=74 |
| Yes | 7 (9) |
| No | 67 (91) |

SD: Standard deviation. ART: Antiretroviral therapy. [a] Financially dependent on his parents.

**Table 2. Infectious history and ART management of the Mexican postpartum WLWH.**

| Characteristic | Mean±*SD* [range] Frequency (%) N=75 |
|---|---|
| **Other chronic diseases** | |
| Any | 6 (8) |
| None | 69 (92) |
| **Perceived HIV transmission route** | |
| Sexual relationships | 64 (85) |
| Needles | 3 (4) |
| Blood transfusion | 3 (4) |
| Vertical | 2 (3) |
| Do not know | 3 (4) |
| **Years since the HIV diagnosis (years)** | 6±5 [0-25] |
| 0–2 | 24 (32) |
| 3–5 | 18 (24) |
| 6–10 | 20 (27) |
| 11–15 | 9 (12) |
| 16–20 | 3 (4) |
| More than 20 | 1 (1) |
| **HIV Symptoms** | |
| Asymptomatic | 73 (97) |
| Symptomatic | 2 (3) |
| | n=62 |
| **Last CD4 count during pregnancy (cells/mm³)** | 533±434 [36-2622] |
| More to 500 | 29 (47) |
| 200–499 | 21 (34) |
| Less to 200 | 12 (19) |
| | n=71 |
| **Initial viral load (copies/mL)** | 484,695±2'453,085 [41−13'000,095] |
| Undetectable | 36 (56) |
| Less to 400 | 6 (9) |
| 401 to 100 mil | 20 (31) |
| More to 100 mil | 2 (3) |
| **Current viral load[a] (copies/mL)** | n=67 |
| Undetectable | 53 (79) |
| Less 400 copies/mL | 7 (10) |
| 401 to 100 mil copies/mL | 6 (9) |
| More 100 mil copies/mL | 1 (1) |
| **Previous HIV-infected children** | |
| Yes | 7 (9) |
| No | 50 (67) |
| N/A | 18 (24) |
| **Current partner serostatus** | n=73 |
| Known | 55 (75) |
| Unknown | 18 (25) |

*(Continued)*

**Table 2.** (Continued)

| Characteristic | Mean±*SD* [range] Frequency (%) N = 75 | |
|---|---|---|
| **ARTs drugs** | **Pregnancy** | **Postpartum** |
| AZT + 3TC + LPV/r | 1 (1) | 1 (1) |
| AZT + 3TC + EFV | 16 (22) | 16 (23) |
| FTC + TDF + LPV/r | 5 (7) | 5 (7) |
| RAL + FTC + TDF | 32 (43) | 32 (46) |
| FTC + TDF + DTG | 6 (8) | 3 (4) |
| BIC + FTC + TAF | 6 (8) | 6 (9) |
| FTC + TDF + DRV + RTV + RAL | 4 (5) | 2 (3) |
| FTC + TDF + ATV + RTV | 1 (1) | 1 (1) |
| ABC + 3TC + EFV | 1 (1) | 1 (1) |
| Other antiretroviral drugs | 2 (3) | 2 (3) |
| **ARTs therapy by group** | | |
| Group A (2 NRTIs + 1 NNRTI) | 17 (23) | 17(25) |
| Group B (2 NRTIs + 1 PI) | 8 (11) | 7 (10) |
| Group C (2 NRTIs + 1 INSTI) | 49 (66) | 45 (65) |
| **Adherence ART (Pregnancy and postpartum)** | | |
| High | 63 (84) | |
| Medium | 5 (7) | |
| Low | 7 (9) | |
| **Intrapartum prophylactic ART** | | |
| Yes | 58 (77) | |
| No | 17 (23) | |
| **Efavirenz during pregnancy** | | |
| Yes | 18 (25) | |
| No | 53 (75) | |
| **Perception of perinatal complications[b]** | n = 74 | |
| Yes | 26 (35) | |
| No | 48 (65) | |

**Nucleoside analog reverse transcriptase inhibitors (NRTIs):** Lamivudine (3TC), Abacavir (ABC), Emtricitabine (FTC), Tenofovir Alafenamide (TAF), Tenofovir Disoproxil Fumarate (TDF), Zidovudine (AZT). **Non-nucleoside analog reverse transcriptase inhibitors (NNRTIs):** Efavirenz (EFV). **Protease Inhibitors (PI):** Darunavir (DRV), Lopinavir/ritonavir (LPV/r), Ritonavir (RTV), Atazanavir (ATV). **Integrase strand transfer inhibitors (INSTI):** Bictegravir (BIC), Dolutegravir (DTG), Raltegravir (RAL). [a]Viral load close to delivery. [b]perception of WLWH about perinatal complications such as high-stakes events, bleeding, prolonged labor, threatened miscarriage, premature rupture of membranes, gestational diabetes mellitus, growth restriction, and the health of the newborn due to prematurity, **neonatal respiratory distress, and others.**

most common ART regimen during prenatal care *vs* postpartum period in all WLWH included two reverse transcriptase inhibitors (NRTIs) plus integrase inhibitors (II) (66 and 65%) categorized as group C (Table 2). During birth, 76% of all participants received ART intrapartum as prophylaxis, based on clinical criteria and in accordance to the national guidelines for the care of WLWH [32]. Based on questionnaires applied during postpartum period, the adherence to ART was high (84%) and confirmed by medical records. One in three women reported perinatal complications. There are no clinical or medical reports related to the use of ART therapy in this study.

## Perception of Quality of life in the postpartum period of the Mexican WLWH

In our study population, the WHOQoL-HIV-Bref instrument demonstrated excellent reliability (Cronbach's index, α = 0.8). The total QoL median score was 15 [IQR:13–16], with the Physical health and Independence level domains scoring highest, 16 [IQR: 13–18 and 14–18, respectively], and the Spirituality, religion and personal beliefs domain scoring lowest, 13 [IQR: 10–16]. Psychological health and Environment median sores were 14 [IQR: 13–17 and 13–16, respectively] (Table 3).

Financial resources, transport, recreation and leisure activities, forgiveness and blame, negative emotions, future concerns, death and dying, and cognitive skills were the sub-domains with the lowest scores (Table 3).

Using as a reference point the median of total QoL score 15 [IQR:13–16], the perception of QoL was categorised as low (<15), medium (15), and high (>15). Despite an overall medium QoL, 49% of the Mexican postpartum WLWH perceived a low total QoL (Fig 1). The results of the S1 Table indicate a statistically significant difference in the total QoL median scores among the ART regimens (Kruskal-Wallis, *p = 0.039*) and in the Physical health domain score (*p = 0.008*). These findings suggest that the treatment regimen is associated with variations in perceived QoL.

Most participants had less than one year since their last obstetric event (73%). Significant differences in median Physical health (Mann-Whitney U, *p = 0.035*) and Social relationships (*p = 0.011*) domain scores were observed across postpartum time groups (S2 Table). Significant differences in median QoL scores across years of interview were observed in the Physical health domain (Kruskal-Wallis, *p < 0.001*), with scores showing a notable decline and reaching their lowest values in 2022. Likewise, significant differences were found in the Social relationships domain (*p < 0.001*) and in Spirituality, religion and personal beliefs (*p = 0.017*) (S2 Table).

## ART is related to physical health, social relationships and overall perception of quality of life in Mexican postpartum WLWH

To determine the relationship between QoL instrument scores with demographic, nonpathological history, infectious history and ART management, obstetric history, and sexual behavior variables a Spearman's correlation test was performed. There was a negative correlation between Physical health and ART therapy during prenatal care (r = −0.313, *p = 0.007*) and postpartum (r = −0.425, *p < 0.001*), and its category (prenatal care r = −0.330, *p = 0.004*; postpartum r = −0.378, *p = 0.001* respectively). Additionally, ART during prenatal care positively correlated with Social relationships (r = 0.260, *p = 0.025*), while ART during postpartum and its category showed positive correlations with this same score (r = 0.294, *p = 0.014*, and r = 0.261, *p = 0.03*, respectively). Lastly, it was observed that ART category during prenatal care and postpartum were negatively correlated to the overall QoL (r = −0.272, *p = 0.019*; r = −0.289, *p = 0.016*, respectively) (Fig 2). This underlies the importance of the role of ART on the perception of QoL of the postpartum WLWH.

Other significant correlations were identified with the QoL scores summarized in Fig 2. In brief, Physical health negatively correlated with Year of interview (r = −0.286, *p = 0.013*), and Efavirenz during pregnancy (r = −0.275, *p = 0.019*). Psychological health positively correlated with Monthly income (r = 0.243, *p = 0.038*), Last CD4 count during pregnancy (r = 0.262, *p = 0.031*) and Age of the beginning of sexual life (r = 0.259, *p = 0.028*), while negatively with HIV symptoms (r = −0.230, *p = 0.047*), Current partner HIV status (r = −0.253, *p = 0.031*), and Tattoos and piercings (r = −0.280, *p = 0.035*). Independence level had a positive relationship with Gestations (r = 0.282, *p = 0.014*), and negative relations with Educational status (r = −0.233, *p = 0.044*), HIV symptoms r = −0.229, *p = 0.048*) and Number of sexual partners (r = −0.417, *p < 0.001*). Social relationships domain had a positive relation with the Year of interview (r = 0.826, *p < 0.001*), while a negative relation with Marital status (r = −0.237, *p = 0.041*), Time of postpartum (r = −0.440, *p < 0.001*), Gestational age at medical admission (r = −0.591, *p < 0.001*), Current partner HIV status (r = −0.260, *p = 0.026*), Caesarean sections (r = −0.230,

**Table 3. WHOQoL-HIV-Bref instrument scores in the Mexican postpartum WLWH.**

| Domains<br>Subdomains | WHOQoL-HIV Bref score<br>transformed 4–20<br>Median [IQR] |
|---|---|
| **Overall QoL and general health** | **16 [14–16]** |
| Quality of Life | 16 [12–16] |
| Health status | 16 [16] |
| **I. Physical health** | **16 [13–18]** |
| Pain and Discomfort | 16 [12–16] |
| Symptoms of WLWH | 16 [12–20] |
| Energy and fatigue | 16 [12–16] |
| Sleep and rest | 16 [12–20] |
| **II. Psychological health** | **14 [13–17]** |
| Positive feelings | 20 [15–20] |
| Cognitive skills | 12 [8–20] |
| Bodily image and appearance | 16 [12–20] |
| Self-steem | 16 [16–20] |
| Negative feelings | 12 [8–16] |
| **III. Independence level** | **16 [14–18]** |
| Dependence of medication or treatments | 20 [12–20] |
| Mobility | 16 [16–20] |
| Activities of daily living | 16 [12–16] |
| Work capacity | 16 [12–20] |
| **IV. Social relationships** | **15 [13–16]** |
| Social inclusion | 16 [16–20] |
| Personal relationships | 16 [12–16] |
| Sexual activity | 16 [12–16] |
| Social support | 16 [12–16] |
| **V. Environment** | **14 [13–16]** |
| Physical safety and security | 16 [12–16] |
| Home environment | 16 [12–16] |
| Financial resources | 12 [8–12] |
| New information and skills | 16 [16–20] |
| Recreation and leisure activities | 12 [8–16] |
| Physical environment | 16 [12–20] |
| Health and social care: accessibility and quality | 16 [16–20] |
| Transport | 12 [12–16] |
| **VI. Spirituality, religion and personal beliefs** | **13 [10–16]** |
| Sense of life (SRPB) | 16 [16–20] |
| Forgiveness and blame | 12 [4–20] |
| Concerns about future | 12 [8–16] |
| Death and dying | 12 [4–20] |
| **Total quality of life (QoL)** | **15 [13–16]** |

*IQR*: [Interquartile range]

 

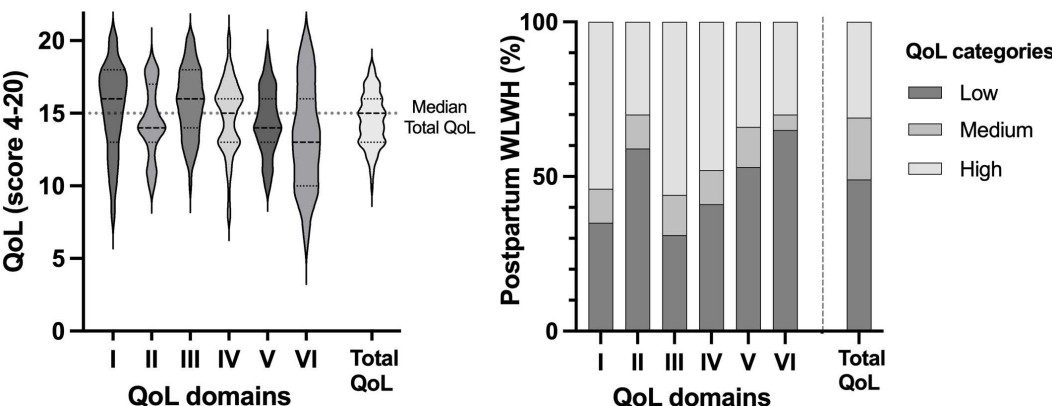

**Fig 1. The Quality of Life (QoL) median scores of the Mexican postpartum WLWH obtained with the WHOQoL-HIV-Bref instrument.** On the left-hand side, scores are presented on a 4–20 points scale. The percentage of postpartum WLWH with the QoL categorized is shown on the right-hand side. Instrument domains are defined as: I. Physical health, II. Psychological health, III. Independence level, IV, Social relationships, V. Environment, VI. Spirituality, religion and personal beliefs. Total QoL refers to the score obtained by the WHOQoL-HIV-Bref instrument. QoL categories: Scores below 15 points are considered low, score of 15 points is considered medium, and scores above 15 are considered high.

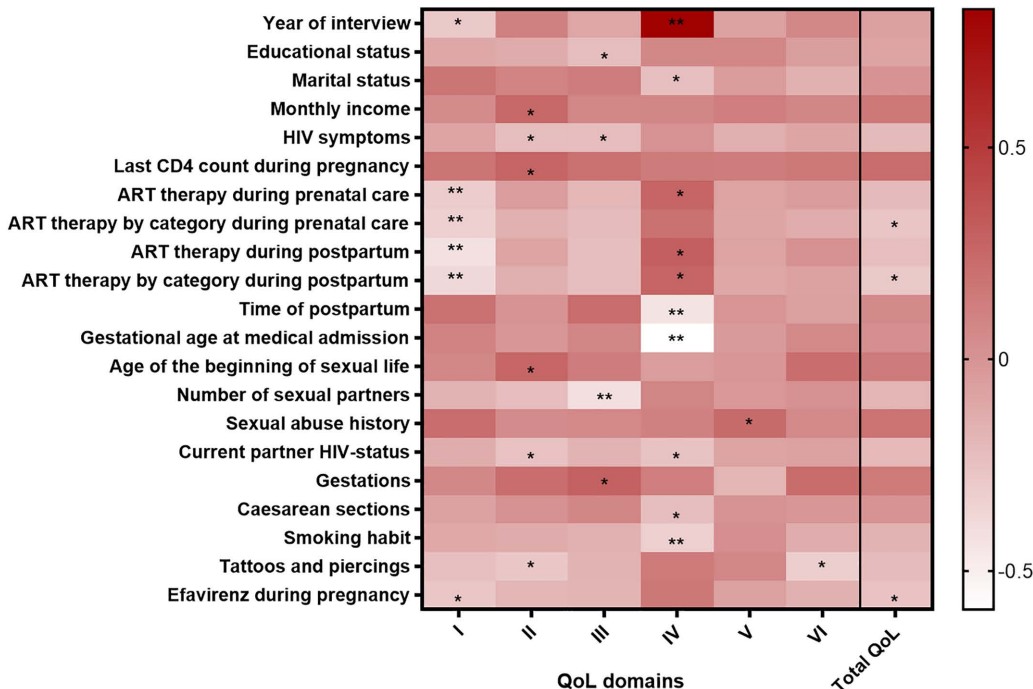

**Fig 2. ART is related to Physical health and Social relationships in the Mexican postpartum WLWH.** Heatmap of the Spearman correlation coefficients between demographic, nonpathological history, infectious history and ART management, obstetric history, and sexual behavior with QoL instrument scores. Instrument domains are defined as: I. Physical health, II. Psychological health, III. Independence level, IV, Social relationships, V. Environment, VI. Spirituality, religion and personal beliefs. Total QoL refers to the score obtained by the WHOQoL-HIV-Bref instrument. A *p-value<0.05* was considered statistically significant.

*p = 0.047*), and Smoking habit (r = −0.325, *p = 0.005*). Lastly, Environment showed a positive relation with Sexual abuse history (r = 0.231, *p = 0.047*), Spirituality, religion and personal beliefs were negatively related to Tattoos, and piercings (r = −0.321, *p = 0.015*), and Total QoL was negatively related to Efavirenz during pregnancy (r = −0.263, *p = 0.025*). Complete correlation coefficients between variables are summarised in S3 Table.

**Physical health, Independence level, and Spirituality, religion and personal beliefs predict a perception of high QoL in Mexican postpartum WLWH**

With the purpose of identifying potential predictors of QoL in the Mexican postpartum WLWH, analyses were conducted as established in the methods. From all the variables tested by MLR and PCA, Age, Years since HIV diagnosis, Gestational age at medical admission, Abortions, Marital status, Domain I. Physical Health, Time of postpartum, Age of the beginning of sexual life, Gestations, Other chronic diseases, Domain III. Independence level, Domain IV. Social relationships, and Domain VI. Spirituality, religion and personal beliefs fulfilled the criteria for further evaluation as potential predictors of QoL. From the above, Domain I. Physical health, Domain III. Independence level, and Domain VI. Spirituality, religion and personal beliefs exhibited good AUC values when tested for high QoL (Fig 3, 0.86, *p = 0*; 0.83, *p = 0*; and 0.84, *p = 0*, respectively). Cutoff values were then calculated for these three domains by the Youden index with their corresponding 95% CI. Summary data regarding these three potential predictors of a perception of high QoL in postpartum WLWH are presented in Table 4.

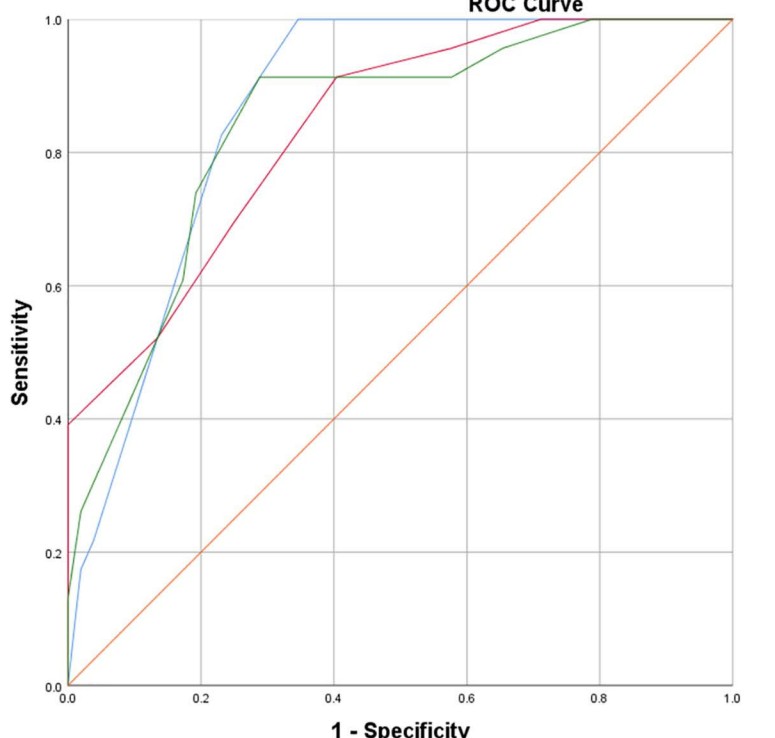

**Fig 3. Physical health, Independence level and Spirituality, religion and personal beliefs may potentially predict a perception of high QoL in Mexican postpartum WLWH.** ROC curve analysis of domains I, III, and VI as markers of a perception of high QoL.

**Table 4. Summary data of potential predictors of a perception of high QoL in the Mexican postpartum WLWH.**

| Domains | AUC | *P* value | Sensitivity | Specificity | Cutoff value | Youden index | 95% CI Lower % | 95% CI Upper % |
|---|---|---|---|---|---|---|---|---|
| I. Physical health | 0.86 | 0.00 | 1.00 | 0.65 | 15.50 | 0.65 | 85.69 | 100.00 |
| III. Independence level | 0.83 | 0.00 | 0.91 | 0.60 | 15.50 | 0.51 | 73.20 | 97.58 |
| VI. Spirituality, religion and personal beliefs | 0.84 | 0.00 | 0.91 | 0.71 | 13.50 | 0.63 | 73.20 | 97.58 |

Data resulted from AUCROC analysis, 95% CI of the Cutoff values was determined by the Wilson Score interval considering a n = 23 patients for the category "high" in QoL. Abbreviations: CI, Confidence Interval; AUC, Area under the curve, *p*-value <0.05.

Additionally, a PCA was performed separately in QoL categories low, medium, and high, to identify which variables accounted for most of the variance per category. Absolute loading values per variable (> 0.6), and the subcategories with the higher frequencies are reported in S4 Table. It is of notice that certain variables were shared between low, medium and high categories: Domain IV. Social Relationships (score of 15 vs 16, low QoL18.9% vs medium 26.7% and high 43.5%), Year of interview (2021 most frequent in high QoL 73.9%, vs medium 66.7% and low 56.8%), Gestations (2 vs 3, low 37.8% and medium 26.7% vs high 34.8%), and c group ART therapy by category during prenatal care (low QoL 78.4% vs medium 57.1% and high 56.5%) and during postpartum (low QoL 76.5% vs medium 53.8% and high 54.5%), suggesting the importance of further exploration of these variables in more extended cohorts.

## Discussion

Notwithstanding, the compelling counsel proffered by numerous experts in the field, the 2030 95-95-95 targets established by UNAIDS notably exclude the 'fourth 95' target, which stipulates that 95% of people living with HIV (PLWH) should attain a QoL that is comparable to that of individuals who are HIV-negative. This target emphasises the identification of factors that reduce the perception of QoL and the establishment of strategies to address them and support progress toward its achievement [4–8].

Our participants received medical care at a tertiary medical institution in Mexico City, most of them were in the third trimester of pregnancy, average 32 gestational weeks (Table 1). In Mexico, delayed prenatal care remains a critical public health concern, disproportionately affecting vulnerable populations, such as WLWH. Disparities are most pronounced among women with lower educational attainment, those residing in rural areas, and individuals of lower socioeconomic status—inequities that were further exacerbated by the COVID-19 pandemic [33,34]. Our findings align with previous studies [24,29,30,35–39], showing that HIV/SIDA disproportionally affects the population in disadvantaged socioeconomic strata. Limited education restricts access to well-paid employment. These women reside in marginalised suburbs of Mexico City, where there is a deficiency in access to adequate, high-quality public transportation at an affordable price. Furthermore, the safety of these areas is compromised by the high crime rates that prevail. Women's reliance on transport systems is fundamental to their economic participation and social integration. However, inadequate accessibility, spatial constraints, and pervasive safety concerns significantly restrict their mobility, producing forms of social exclusion. The absence of secure and reliable travel options not only shapes women's perceptions of the urban public realm but also curtails their access to education, employment, and healthcare. Consequently, mobility and safety emerge as critical determinants of gender equity and broader societal inclusion (Table 1).

The significant differences in total QoL scores and in the Physical health domain across ART regimens suggest that treatment choice may influence patients' perceived well-being. Most participants received ART regimen consisting of two nucleoside reverse transcriptase inhibitors (NRTIs) and one integrase inhibitor (II), classified as group C. The predominance of this regimen reflects current clinical guidelines that prioritize II-based combinations due to their high efficacy, favourable resistance profile, and generally good tolerability [32]. During their pregnancy they perceived high adherence,

which persisted until postpartum, as reported in their clinical files. Many also received intrapartum prophylactic ARTs according with the national guidelines for the care of WLWH [32], and underwent Caesarean sections (Table 2). Studies indicate a high adherence to ART during pregnancy, which decreases after delivery [36]. Pregnant WLWH prioritise adherence to prevent vertical transmission and ensure their health to care for their babies [37]. Most of the participants had a recent diagnosis, which may help patients remain physically well and focus on their ART, despite psychological distress [11].

In our sample, 75% of the participants are aware of their partners' HIV status. It should be noted that some of these individuals are HIV-positive (Table 2) and disclosed their diagnosis only to current partners or close family, no extended relatives. This behaviour coincides with previous studies, where PLWH only tell their partners or immediate family members about their diagnosis. Extended family members, such as parents-in-law, siblings, and grandparents, are not informed due to stigmatization [25,40]. Parental support from partners during pregnancy has been linked to better adherence to ART [41].

The QoL instrument demonstrated excellent internal consistency, aligning with evidence from the English version validated in Ethiopia [30,38], as well as the Spanish version [28].

The group of Mexican postpartum WLWH in this study had an intermediate total QoL median score (Table 3 **and** Fig 1) despite belonging to the lowest socioeconomic strata. Their undetectable viral load, adequate CD4 cell count, and adherence to ART contributed to an optimistic perception and HIV-free births. Nevertheless, the educational status, HIV-related symptoms, partner's serostatus, and histories of sexual abuse intensify vulnerability, stigma and intersect as indicators of higher risk behavior impacting both mental and physical health; while treatment status, regimen type, and limited income further constrain coping capacity, influencing the perception of QoL (Fig 2). The above reflects the impact of socioeconomic constraints despite the high Human Development Index (HDI) of Mexico (HDI, 0.781) [42]. Vulnerable groups face limited access to essential services, lowering QoL scores [28]. A study in India [35], found higher QoL scores, among non-pregnant-WLWH, despite the poverty. The study attributes this result to a structured program for the distribution of ART, close monitoring of adherence, and strong community support. It is important to note that India has a lower HDI (0.644) compared with Mexico. In contrast, in Ethiopia (HDI, 0.492), Iran (HDI, 0.780), Brazil (HDI, 0.760), and even Spain (HDI, 0.911) [42], limited education, lack of community support, barriers to ART access due to gender, socioeconomic inequalities, stigma, and racism were related to low QoL, poor adherence, and severe health problems [24,28–30]. We suggest that HDI is not associated with the perception of QoL.

The total QoL comprises six domains (Fig 1) with global differences most evident in median scores. Our participants reported energy for daily tasks (Physical and Independence level domains) and satisfaction with their relationships due to partners, friends, or family support (Relationship domain), but are constrained by marital status, postpartum duration, clinical conditions. The findings indicate that postpartum duration was a significant factor affecting women's QoL. The duration since childbirth has been shown to impact women's perceived social support and relational well-being, with a time frame over one year being particularly notable. On the other hand, when the value is less than one year, the Physical health domain receives a low score [43,44]. The stigma surrounding their condition often led to secrecy, anxiety, and reduced QoL scores in the domains of psychological health and personal beliefs. According to a study in northern Mexico, stigma and a lack of societal acceptance of people living with HIV/AIDS (PLWH) discourage them from disclosing their status to family, friends, and coworkers [45].

Our postpartum WLWH expressed dissatisfaction with their living conditions due to insufficient services and transportation, as they live in the suburbs of Mexico City, in addition to financial constraints that limit access to health care and leisure activities, as reflected in a low Environment domain score (Table 3). Moreover, their wages were lower than men's when they were employed; likewise, the low psychological health scores highlight ongoing stigma and discrimination, with perception of judgment tied to forgiveness and guilt, as reported in Spain [28], Ethiopia [30,38], Iran [39], Brazil [24], and India [35]. Also, we could say that the QoL perceptions varied according to cultural, traditional, and religious

factors. In this respect, Haraldstat et al. (2019) concluded that the concept of QoL may vary across cultures, with unclear cross-cultural relevance. Further studies in Asian and non-Western cultures are needed to explore QoL and its cultural manifestations [18].

The present study also provides insight into the impact of the ART therapy on QoL domains. The evidence presented herein demonstrates the dual impact of ART during pregnancy and postpartum of WLWH. The negative correlation with the physical health domain suggests that treatment side effects or the burden of the treatment regimen may reduce perceived well-being. In contrast, the positive correlation with the Social relationships domain indicates that ART use enhances feelings of safety and strengthens interpersonal support. These findings underscore the importance of comprehensive care approaches that address physical discomfort while also fostering social inclusion in WLWH (Fig 2). These findings align with the conclusions of previous studies, which have indicated that women receiving long-term ART can expect to maintain a satisfactory QoL [35,38]. However, it is possible that specific antiretroviral regimens may have exerted a negative influence on outcomes. This is exemplified by the utilisation of efavirenz during pregnancy (Fig 2). As Xiao et al. (2023) and Law et al. (2020) have demonstrated, this antiretroviral drug has exerted a substantial deleterious effect on the Physical health domain. This is attributable to the fact that such agents have been associated with the occurrence of neuropsychiatric adverse events in PLWH, which frequently result in a diminution of QoL and a consequent decline in patient adherence [46,47].

Stressors such as the COVID-19 pandemic, which disrupted healthcare access and daily routines. The significant differences observed in Social relationships and Spirituality, religion and personal beliefs domains (S2 Table) further suggest that Psychosocial dimensions of well-being were negatively affected, possibly due to restrictions on social interaction and reduced opportunities for community or spiritual support during this period [48,49]. Physical health domain was also negatively associated with year of interview suggesting pandemic-related influences on well-being (Fig 2 **and** S2 Table).

The global population experienced significant impacts from the SARS-CoV-2 (COVID-19) pandemic, including economic, social, psychological, environmental, and public health challenges. The disease posed life-threatening risks and immense psychological stress worldwide. The relationship between the perception of vulnerability (e.g., unemployment, food, job insecurity, and illness) and low QoL is widely accepted [50]. Teotonio *et al*. reported that females presented worse QoL than males, due to predominantly manage family food purchases, meal preparation, and household decisions. During social distancing, women bore most domestic chores, childcare, education, and professional responsibilities, adversely affecting their QoL [50]. Mexican postpartum WLWH were not the exception, they had reduced scores in the physical health domain in 2022 compared to 2020–2021 scores (S2 Table). In accordance with other studies [50–56] also the increased stress, isolation, gender inequities, and reduced physical activity during lockdown, exacerbated vulnerability and wellness perceptions among WLWH.

For WLWH, the post-lockdown era signifies more than a mere return to normality during pregnancy and postpartum. The situation is still significantly impacted by a combination of ongoing psychosocial, structural, and HIV-related factors, and efforts have been made to address their emotional and psychological well-being and persistent stigma [43,44,48].

The correlations across QoL domains (Fig 2) highlight the interplay of medical, social, and cultural factors in WLWH during the postpartum period. As demonstrated in preceding studies, the consequences primarily encompass physical health, psychological well-being, independence levels, and relationships [24,25,28–30,35,38,39,40,41,57–65].

The principal component analysis (PCA) across low, medium, or high QoL categories revealed that several variables consistently contributed to variance, underscoring their relevance as potential determinants of the perception of QoL in Mexican postpartum WLWH. Social relationships domain emerged as a shared factor across all categories, highlighting the central role of interpersonal support in shaping well-being. Evidence shows that social support fosters psychological well-being, self-esteem, and resilience, translating into healthier coping and reduced stress responses [63]. Similarly, the year of the interview suggests that temporal or contextual factors, such as evolving health policies or social support structures, may have influenced QoL outcomes. This was indeed the case, as the study was conducted between 2020 and

2022—the years of the SARS-CoV-2 pandemic and the lockdown— with the consequences for this population previously described [50–56]. Furthermore, the gestational history also appeared across categories, reflecting the influence of maternal experience on resilience and self-perceived QoL. The notion that repeated maternal experiences may enhance self-reliance and confidence in managing daily life has been postulated by Puente Ferreira and Hernando (2023) and Lima et al. (2022). These researchers concluded that the experience of motherhood can boost the confidence and self-assurance of women living with HIV, allowing them to reaffirm their feminine identity, normalise their lives and overcome the social stigma associated with the disease. The advent of medical advances has rendered it feasible for women to have healthy children, thereby empowering them to prioritise their own well-being and demonstrate a commitment to their health [62,66]. The use of ART therapy during prenatal and postpartum care was more prevalent among women with low QoL scores, suggesting that treatment regimens may contribute to differences in perceived health and functioning. ART is an essential component of viral suppression; however, it can also contribute to adverse effects or treatment-related discomfort, which can negatively influence the perceived physical well-being of these women. It is important to note that certain antiretroviral regimens may have had a detrimental effect on physical outcomes, and this discomfort can impede WLWH participation in social activities. The aforementioned points are in accordance with previous studies [8,35,37,38,46,47]. These findings emphasize the need for further exploration of these variables in larger cohorts to clarify their predictive value and inform targeted clinical interventions.

The multinomial linear regression (MLR analysis) identified several demographics, clinical, and psychosocial variables as potential predictors of QoL of WLWH in postpartum. In the present study, three domains—Physical health, Independence level, and Spirituality—were identified as the strongest predictors of elevated QoL scores. Each domain demonstrated a high discriminative capacity (Fig 3). The predictive value of the Physical Health domain emphasises the significance of effective symptom management and treatment-related adverse effects in determining overall well-being. The Independence domain is indicative of the role of self-reliance and functional autonomy, both of which are critical for coping with the demands of postpartum life. Finally, the domain Spirituality, religion, and personal beliefs demonstrate the protective influence of cultural and existential frameworks in sustaining resilience. The aforementioned supports the results here obtained and are consistent with previous studies [8,35,38,40,46,47,61,64,66–73], which together with these findings emphasize the necessity of comprehensive interventions that integrate clinical care with psychosocial and cultural dimensions to increase QoL in this population.

No single instrument can fully evaluate all QoL factors. WHOQoL-HIV-Bref, although robust, does not fully assess barriers, such as mental illness and addictions or specific aspects related to ART (e.g., adherence, duration and type). More research should use specialised tools to assess adherence to ART and mental health. It is important to stablish that this instrument assesses the QoL and includes two questions about self-perceived QoL and health status, which score we called overall QoL and general health (Table 3).

This study had several limitations. First, obtaining a large sample of postpartum WLWH was challenging due to the specificity of their characteristics and vulnerability. Second, the cross-sectional design prevented the establishment of causality between variable associations. Third, ART adherence was obtained through clinical interviews rather than specialized instruments. Fourth, perception and behavior of QoL may have been influenced by the lockdown and post-pandemic period of the SARS-CoV-2 (COVID-19) pandemic.

Despite its limitations, this study provides valuable information on highly vulnerable populations such as postpartum WLWH and their QoL scores. The sample size is representative of Mexican postpartum WLWH who were treated at the antenatal care at INPer. The participants are from Mexico City and its suburbs (the Estado de México).

Nevertheless, the analysis conducted enabled the identification of significant sociodemographic and clinical variables that contribute to the assessment of QoL, and the identification of three predictors of a high QoL. The emphasis is on the identification and mitigation factors influencing the perception of QoL in PLWH, specially in high vulnerability groups as a WLWH during pregnancy and postpartum, with the aim of proposing strategies for medical attention and therapeutic

success, and for the promotion of overall well-being. Furthermore, the call is made for the inclusion of QoL evaluation in WHO strategies, the fourth 95 target, and government and societal efforts to reduce stigma, foster acceptance, and improve environments for WLWH, future generations, and exposed newborns.

## Conclusions

HIV/AIDS has a detrimental impact on the perception of QoL of Mexican postpartum WLWH and potentially their infants. Despite poverty, stigma, and discrimination, they remain motivated, especially when their child is born HIV-free. Mexican postpartum WLWH perceived their QoL as moderate, largely influenced by dual impact of ART regimens during pregnancy and postpartum, along with low-income. Clinical factors, particularly undetectable viral load, obstetric history, and the absence of symptoms due to HIV infection, have been found to be closely linked to improve their perception of QoL. The domains Physical health, Independence level, and Spirituality, religion and personal beliefs scores are strongest predictors of elevated total QoL scores. These findings underscore the significance of formally incorporating QoL assessment as the 'fourth 95' target in WHO strategies for the most vulnerable among PLWH, WLWH and their children.

## Supporting information

**S1 File. WHOQoL-HIV-Bref instrument (Adapted version).**
(DOCX)

**S1 Table. Comparison of the total QoL median scores and their domains in the Mexican postpartum WLWH using three different combinations of ART.**
(DOCX)

**S2 Table. Comparison of the total QoL median score and their domains at different postpartum times and in different years of interviews of the Mexican postpartum WLWH.**
(DOCX)

**S3 Table. Spearman's correlation coefficients between demographic and non-pathological history, infectious history, ART management, obstetric history and sexual behavior *vs* total QoL score and their domains in the Mexican postpartum WLWH.**
(DOCX)

**S4 Table. Results of Principal Component Analysis (PCA): Variables and their categories that account for the most total variance in classification into low, medium or high QoL.**
(DOCX)

## Acknowledgments

The authors are also deeply grateful to all the women who participated in this work, and we hope that our efforts will improve their quality of life in all respects. The authors thank to their colleagues, Estela Y. Godínez-Martínez, Virginia E. Santillán-Palomo, María del Pilar Meza-Rodríguez, Carmen Flores-Cisneros, Cecilia Mota-González, Rosa Georgina Rodríguez-Delgado, and Edgar Bonilla-Reyes, for their excellent technical assistance, interest in this work, and helpful discussions.

## Author contributions

**Conceptualization:** Jessica Hernández-Pineda.

**Data curation:** Mauricio Domínguez-Castro, Alicia Ramírez-Ramírez, Miroslava Avila-García, Edna Basilio-Gálvez, Jessica Hernández-Pineda.

**Formal analysis:** Mauricio Domínguez-Castro, Alicia Ramírez-Ramírez, Miroslava Avila-García, Edna Basilio-Gálvez, Margarita López-Martínez, Ismael Mancilla-Herrera, Jessica Hernández-Pineda.

**Investigation:** Mauricio Domínguez-Castro, Alicia Ramírez-Ramírez, Noemí Guadalupe Plazola-Camacho, Miroslava Avila-García, José Romo-Yáñez, Carmen Selene García-Romero, Jessica Hernández-Pineda.

**Methodology:** Mauricio Domínguez-Castro, Alicia Ramírez-Ramírez, Noemí Guadalupe Plazola-Camacho, Edna Basilio-Gálvez, Margarita López-Martínez, Ismael Mancilla-Herrera, Jessica Hernández-Pineda.

**Project administration:** Noemí Guadalupe Plazola-Camacho, Diana Mercedes Soriano-Becerril.

**Software:** Edna Basilio-Gálvez.

**Supervision:** Mauricio Domínguez-Castro, Alicia Ramírez-Ramírez, Noemí Guadalupe Plazola-Camacho, Norah Lucky Katende-Kyenda, José Romo-Yáñez, Jessica Hernández-Pineda.

**Validation:** Mauricio Domínguez-Castro, Alicia Ramírez-Ramírez, Miroslava Avila-García, Margarita López-Martínez, Norah Lucky Katende-Kyenda, Diana Mercedes Soriano-Becerril, Carmen Selene García-Romero, Ricardo Figueroa-Damián, Jessica Hernández-Pineda.

**Visualization:** Margarita López-Martínez, Ismael Mancilla-Herrera, Norah Lucky Katende-Kyenda, Diana Mercedes Soriano-Becerril, José Romo-Yáñez, Ricardo Figueroa-Damián, Jessica Hernández-Pineda.

**Writing– original draft:** Mauricio Domínguez-Castro, Alicia Ramírez-Ramírez, Miroslava Avila-García, Edna Basilio-Gálvez, Jessica Hernández-Pineda.

**Writing– review & editing:** Mauricio Domínguez-Castro, Alicia Ramírez-Ramírez, Noemí Guadalupe Plazola-Camacho, Miroslava Avila-García, Margarita López-Martínez, Ismael Mancilla-Herrera, Norah Lucky Katende-Kyenda, Diana Mercedes Soriano-Becerril, José Romo-Yáñez, Carmen Selene García-Romero, Ricardo Figueroa-Damián, Jessica Hernández-Pineda.

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
