## [Decision Letter · Decision Letter 0]

16 Jan 2026

PONE-D-25-40652Quality of life associated with clinical and sociodemographic determinants in the postpartum period of Mexican women living with HIV. A cross-sectional studyPLOS One Dear Dr. Hernández-Pineda,

Thank you for submitting your manuscript to PLOS ONE. After careful consideration, we feel that it has merit but does not fully meet PLOS ONE’s publication criteria as it currently stands. Therefore, we invite you to submit a revised version of the manuscript that addresses the points raised during the review process.

**ACADEMIC EDITOR: Major revision**==============================

We look forward to receiving your revised manuscript.

Kind regards,

Marwan Salih Al-Nimer, MD, PhD

Academic Editor

PLOS One

Journal Requirements:

This study was supported by the Instituto Nacional de Perinatología (Mexico City) under approval number 212250-3120771.

The authors are also deeply grateful to all the women who participated in this work, and we hope that our efforts will improve their quality of life in all respects. The authors thank to INPer (approval number 212250-3120771), as well as their colleagues, Estela Y. Godínez-Martínez, Virginia E. Santillán-Palomo, María del Pilar Meza-Rodríguez, Carmen Flores-Cisneros, Cecilia Mota-González, Rosa Georgina Rodríguez-Delgado, and Edgar Bonilla-Reyes, for their excellent technical assistance, interest in this work, and helpful discussions.

This study was supported by the Instituto Nacional de Perinatología (Mexico City) under approval number 212250-3120771.

6. Thank you for providing your underlying data as Supporting Information.

We note that the data set contains text or data that is not in English. Please note that PLOS is an English-language publisher, so we require data sets to be provided in English as well. Please upload an English-language version of your data set.

This will also allow us to determine if your data follows PLOS standards per our Data Availability policy here: https://journals.plos.org/plosone/s/data-availability

Additional Editor Comments:

The authors asked to take considerations to the comments of the authors.

Reviewers' comments:

Reviewer's Responses to Questions

**Comments to the Author**

1. Is the manuscript technically sound, and do the data support the conclusions?

Reviewer #1: Partly

Reviewer #2: Yes

Reviewer #3: Partly

2. Has the statistical analysis been performed appropriately and rigorously? 

Reviewer #1: No

Reviewer #2: Yes

Reviewer #3: Yes

3. Have the authors made all data underlying the findings in their manuscript fully available?

Reviewer #1: Yes

Reviewer #2: Yes

Reviewer #3: Yes

4. Is the manuscript presented in an intelligible fashion and written in standard English?

Reviewer #1: Yes

Reviewer #2: Yes

Reviewer #3: No

5. Review Comments to the Author

Reviewer #1: Dr. Suad Ghaben

PONE-D-25-40652

Quality of life associated with clinical and sociodemographic determinants in the postpartum period of Mexican women living with HIV.

Dear Authors

Please find my comments on your manuscript below. This research explored the Quality of Life among pregnant women diagnosed with HIV in the postpartum period. It also identified the correlation of QoL with SD factors among the target population. Your results showed an overall medium QoL, with the psychological domain scoring lowest and physical health and independence scoring highest. About half reported poor QoL, influenced by socio-demographic factors and CD4 counts.

Title:”

• Short, concise, but misleading. The sentence “Quality of life associated with clinical and sociodemographic” judges the relationship. Please correct per the recommendation provided.

• Suggestions: “Predictors of the QoL among the postpartum period of women diagnosed with HIV”

Abstract:

• Follow the PLOS ONE preferred headings of the Abstract.

• Should be revised per the corrected introduction, methods, and results.

• Write the conclusion instead of the discussion.

Introduction:

• The rationale is not clear. What is the progress of QoL? Due to what intervention? Has this intervention been considered in this study?

Methods and materials

The methods section lacks essential reporting of the sampling techniques and the ART intervention, as follows:

• The study design is not appropriately reported. With insufficient explanation of the ART intervention.

• How was sampling conducted? Is it purposive, convenient, or what?

• The inclusion criteria are not clear? Why was ART treatment included? Have you applied any measures to monitor and control the ART intervention?

• Please justify why the sampling was performed during pregnancy, and data collection was performed after delivery.

• How was the ART intervention tracked?

• Have you maintained follow-up of the women from recruitment to the actual data collection? if yes, how was the follow-up maintained?

• What are the clinical determinants that have been evaluated for association with QoL?

• Wrong selection of the statistical test. With multiple Independent variables and one dependent variable, it is advised to use multiple linear regression analysis rather than Spearman’s correlation. In this case, as the QoL is categorized in nominal variables, you should apply the Multiple Regression Model.

Results:

• Page 11: lines 203-204: “32% of our population were diagnosed with HIV during their current pregnancy, and most of the cases they believed that it was sexually transmitted Please justify the prolonged data collection period, about 18 months, considering the small sample size. This might induce balance or lack validity of the data.” Please justify that the study targets women with HIV receiving ART treatment, and the 32% of the sample were diagnosed with HIV?

• S2 file, Table. Justify collecting the QoL scores at different postpartum times of postpartum-WLWH.

Reviewer #2: The manuscript is well written in standard English and required format.

The minimum sample size calculated is 42, how the recruitment of participant is 75 was not explained.

The results/table shows n=75, with many variables showing varying sample size, indicating varying missing data that may have been cleaned before the analysis.

Table 4 (QoL in the women) may be presented as a figure (Pie or Bar Chart) instead as a table.

The discussion, and recommendation are in line with the results presented.

Also, the ethical and confidentiality considerations were stated.

Reviewer #3: Overall: Valuable insights on the QoL of women in the postpartum period, contributing to the growing body of knowledge around this subject.

Feedback

1. The introduction is written in a complicated manner; the remainder of the manuscript is more clearly written and easier to understand (compared to the introduction).

2. Line by line feedback:

3. Line 41:

a. Where would the authors propose that the QOL questionnaire be included in the healthcare system, i.e. where in the postpartum care algorithm or even the antepartum system that the QOL questionnaire be included? Is there a plan to share the findings with health authorities?

b. The authors are cautioned that this is a small sample size and generalizability of results would be limited. Is there a plan to study this area with a larger sample?

4. Lines 59- 60: The link between stigma and discrimination and chronicity of infection is unclear. Perhaps the authors could rather frame the stigma and discrimination as an additional challenge.

5. Lines 61- 67: This sentence is very difficult to follow, and the authors are advised to replace the one sentence with 2-3 linked sentences.

6. Lines 75- 79: This sentence should precede the previous 2 sentences for ease of reading.

7. Line 80- "Affects" may not be the correct word for what the reviewer understands the authors wish to convey- would "encompass" not be a better word?

8. Line 82: Review the document for TENSE- this should be “has.”

9. Lines 90- 94: Please consider simplifying the sentence extending from.

10. Line 95: Incorrect spelling ("focussing").

11. Lines 103- 107: difficult to follow, please simplify.

12. Line 116: what is meant by "this stage"?

13. Line 119: this is an unusually high LTFU- is this based on previous LTFU rates when working with this particular group of women?

14. Line 120: Please advise why 75 women were recruited if the final sample size was calculated as 42.

15. Line 126: please elaborate on how the participants’ anonymity and privacy were protected

16. Line 149: Please confirm the reference for the classification of participant perception of QoL as poor, medium and high.

17. Line 150: Please advise on the reasons for modification of the instrument and if modifications were made to all 11 questions. Further, please also advise on the total number of questions following modifications.

18. Lines 152- 153: the sentence is incomplete.

19. Line 178: Please consider discussing the reasons for late antenatal presentation discussed, especially if linked to QoL.

20. Line 193: slightly misleading statement- might be clearer if the sentence read as "reported smoking (4%) etc" as in its current phrasing it reads as 17% denied knowledge and then admitted to smoking/ alcohol etc in their partners.

21. Line 204: the word "they" is unnecessary.

22. Line 209: Is this only for the newly diagnosed women or for all participants?

23. Line 212: How were the 3 categories of adherence determined?

24. Line 212: Were these complications in the perinatal period related to the ART regimens?

25. Lines 236- 238: Is this an interpretation of the results in table 3?

26. Line 252: please review the sentence construction for the sentence beginning "There were not..."

27. Line 253: please clarify which group of postpartum women is being referred to

28. Line 307: The word "officially" implies that it is endorsed but not yet included in the targets. This 4th target, however, is still under debate and perhaps the authors should consider rephrasing this to indicate that some experts have advised a fourth target etc

29. Line 316: With 92% living in urban areas, please confirm that there is a lack of transport in these areas.

30. Line 316: Please advise how the lack of transport and security "results in discrimination, gender and social inequality and economic dependence."

31. Line 319: Please advise what is referred to by the word "which thrived." Is it adherence that is being referred to? How was this recorded in their clinical files?

32. Line 326: The authors should consider rephrasing this as 47% were aware of their partners serostatus and 26% were unaware of their status. In this 26%, some partners may be seropositive so it is advised that the authors be careful in how they present this information.

33. Line 332: Perhaps the authors would like to consider combining these sentences.

34. Line 343: The part of the sentence starting with "considering that India..." does not link well with the first part of the sentence. Please review.

35. Line 351: Please review the sentence starting with "And"- perhaps this is not meant to be a capital letter?

36. Lines 373- 385: The reviewer acknowledges the evidence presented in relation to the SARS-CoV-2 pandemic. Perhaps the authors could explore the QoL scores post 2020 and if there was improvement, to compare with other studies that might have assessed this.

37. Line 400- 403: The reviewer advises the authors to be cautious in this statement- the sample size is small and specific to one institution.

38. Line 405: consider reworking this sentence to read better.

39. Feedback on Tables included in the manuscript:

a. In Table 1:

i. Under monthly income- please confirm that the first line is meant to read "Depends on his parents." The table presents characteristics of the women on study.

ii. Please advise on the relevance of sexual debut, number of sexual partners, sexual abuse history and tattoos in the context of this study.

b. Table 2:

i. Was VL analysed in relation to duration of ART? 68% of participants have been diagnosed for more than 3 years.

ii. Please clarify why 17 women did not receive intrapartum ART- were they already on treatment? Why were 77 women eligible for intrapartum ART with 68% diagnosed with HIV infection 3 or more years prior?

iii. What does the perception of perinatal conditions refer to?

c. The heading of table 4 is incorrect as all women are included in the table, not only the women with poor QOL.

6. PLOS authors have the option to publish the peer review history of their article (what does this mean?). If published, this will include your full peer review and any attached files.

Reviewer #1: **Yes:** Suad J Ghaben

Reviewer #2: **Yes:** Dr Tawakalit Olubukola Salam

Reviewer #3: No

---

## [Author Response · Author response to Decision Letter 1]

4 Apr 2026

March 17th, 2026.

Marwan Salih Al-Nimer, MD, PhD

Academic Editor

PLOS One

Dear Editor and Reviewers,

We would like to express our sincere gratitude to the reviewers for their valuable comments on the manuscript entitled “Quality of life in the postpartum period of mexican women living with HIV: The role of clinical and sociodemographic factors” by Mauricio Domínguez-Castro and Alicia Ramírez-Ramírez, et al., as the first authors. Your comments were all very helpful in improving the manuscript, and we have incorporated your suggestions into our revision. We have responded to each comment raised by the academic editor and reviewers and addressed them accordingly.

According to journal requirements, we request that you to modify the financial disclosure at point 2. Also, we request modifications in the authors’ list to add PhD. Edna Basilio-Gálvez; her contribution in the advanced statistical analysis was fundamental for reaching the objectives requested by reviewers.

We hope our manuscript is now suitable for publication in the PLOS One.

We look forward to hearing from you at your earliest convenience.

Sincerely yours

MSc. Jessica Hernández Pineda

Department of Infectology and Immunology

National Institute of Perinatology

Phone:+015555209900, ext.497

jesspinq@yahoo.com.mx

---- Journal Requirements ----

Answer. According to your comments, we have reviewed and ensured that this revised version of our manuscript meets PLOS ONE´s requirements.

“This study was supported by the Instituto Nacional de Perinatología (Mexico City) under approval number 212250-3120771.”

Please state what role the funders took in the study. If the funders had no role, please state: "The funders had no role in study design, data collection and analysis, decision to publish, or preparation of the manuscript." Please include this amended Role of Funder statement in your cover letter; we will change the online submission form on your behalf.

Answer. We wish to include the following in the financial disclosure as follows:

This study was supported by the Instituto Nacional de Perinatología (Mexico City) under approval number 212250-3120771. The founders had no role in study design, data collection and analysis, decision to publish, or preparation of the manuscript.

We would appreciate the change in the online submission form.

“The authors are also deeply grateful to all the women who participated in this work, and we hope that our efforts will improve their quality of life in all respects. The authors thank to INPer (approval number 212250-3120771), as well as their colleagues, Estela Y. Godínez-Martínez, Virginia E. Santillán-Palomo, María del Pilar Meza-Rodríguez, Carmen Flores-Cisneros, Cecilia Mota-González, Rosa Georgina Rodríguez-Delgado, and Edgar Bonilla-Reyes, for their excellent technical assistance, interest in this work, and helpful discussions.”

We note that you have provided funding information that is not currently declared in your Funding Statement. However, funding information should not appear in the Acknowledgments section or other areas of your manuscript. We will only publish funding information present in the Funding Statement section of the online submission form. Please remove any funding-related text from the manuscript and let us know how you would like to update your Funding Statement. Currently, your Funding Statement reads as follows: This study was supported by the Instituto Nacional de Perinatología (Mexico City) under approval number 212250-3120771.

Answer. We removed any funding-related text from the manuscript. About the Funder Statement, we would like to update as follows:

“This study was supported by the Instituto Nacional de Perinatología (Mexico City) under approval number 212250-3120771. The founders had no role in study design, data collection and analysis, decision to publish, or preparation of the manuscript.”

Answer. The ethics statement has been removed; it should appear only in the Methods section.

Answer. We appreciate the recommendation. Due to this, we are considering making the dataset available on a repository called Zenodo (https://zenodo.org/), once we submit the revised version back to PLOS One. A dataset uploaded to Zenodo is usually available almost immediately, and if a ‘hold’ period is set, the metadata will be made public immediately, but the files cannot be downloaded until the date you specify.

6. Thank you for providing your underlying data as Supporting Information.

We note that the data set contains text or data that is not in English. Please note that PLOS is an English-language publisher, so we require data sets to be provided in English as well. Please upload an English-language version of your data set.

This will also allow us to determine if your data follows PLOS standards per our Data Availability policy here: https://journals.plos.org/plosone/s/data-availability

Answer: Thank you for your feedback. We apologize for our carelessness and can confirm that the entire manuscript and all supporting information are now in English.

Answer. Your suggestion has been considered in the revised version of the manuscript.

---- Reviewer 1 ----

• TITLE.

Short, concise, but misleading. The sentence “Quality of life associated with clinical and sociodemographic” judges the relationship. Please correct per the recommendation provided.

Suggestions: “Predictors of the QoL among the postpartum period of women diagnosed with HIV”

Answer. We would like to express our sincere appreciation for the reviewer’s suggestion regarding the title. In our study, the objective was to examine the associations and impact of clinical and sociodemographic variables on quality of life, rather than establish predictive models. However, we recognise the rationale behind using the term 'predictors' –which included predictive analyses and their validation as predictors–, and referring to these variables as 'predictors' could misrepresent the scope and methodology. Therefore, to address both approaches, we included predictive analyses and revised the title to encompass both objectives of our study.

Regarding the term ‘women diagnosed with HIV’, since the mid-1990s, the term 'people living with HIV' has been used to distinguish chronic infection from AIDS (the advanced stage), reduce stigma and humanise patients, following the expansion of effective antiretroviral therapy. The terms 'infected' and 'sick' are considered stigmatising and technically inaccurate, as individuals receiving treatment may never develop AIDS. For this reason, we consider 'women living with HIV' to be a more precise and inclusive description.

In light of these considerations, we have revised the title to:

"Quality of life in the postpartum period of Mexican women living with HIV: The role of clinical and sociodemographic factors"

We believe this formulation better reflects the aims of our study, maintains academic neutrality, and provides a more accurate representation of the study population.

• ABSTRACT Follow the Plos One preferred headings of the Abstract.

• Should be revised per the corrected introduction, methods, and results.

• Write the conclusion instead of the discussion.

Answer. We have reorganised the abstract into the following sections: Introduction, Methods, Results, and Conclusions. The statements have been adapted accordingly to fit this structure. We also addressed the suggestion to replace the discussion with a clear conclusion. We hope you agree that this restructuring enhances clarity and aligns with the journal’s requirements.

• INTRODUCTION

• The rationale is not clear. What is the progress of QoL? Due to what intervention? Has this intervention been considered in this study?

Answer. In response to your initial comment that 'the reasoning is unclear', we have carefully revised the manuscript, rewriting several paragraphs in this section to improve clarity and coherence. We hope that these changes make the rationale of our study clearer. Please review from lines: 56-99.

Concerning the question 'What is the progress of quality of life? We interpret this as reflecting the impression that our study is longitudinal, involving repeated application of the QoL instrument (e.g. at baseline and post-intervention). However, we would like to clarify that this is not the case; our research is a cross-sectional study in which the QoL instrument is applied only once. No intervention is planned to compare pre- and post-intervention outcomes. Our objective is to quantitatively assess participants' perceptions of their well-being, and then to explore the potential impact of sociodemographic and clinical factors on these perceptions. In this way, our study aims to provide evidence that supplements the traditional clinical perspective by incorporating less visible yet equally important aspects of well-being.

• METHODS AND MATERIALS

The methods section lacks essential reporting of the sampling techniques and the ART intervention, as follows:

• The study design is not appropriately reported. With insufficient explanation of the ART intervention.

Answer. Regarding the description of the study design and explanation of ART, we would like to clarify that our research was an observational, cross-sectional study. The study did not involve any intervention beyond the standard of care by the institution. As all participants received ART as part of routine care, it was considered a determining factor to analyse for its impact on quality of life, rather than an intervention.

To avoid ambiguity, we have strengthened the methodological description to explicitly state that the study is cross-sectional, with a single application of the QoL instrument. No intervention or longitudinal follow-up was performed. ART forms part of the standard treatment received by all participants and is analysed as a clinical variable possibly associated with QoL rating.

The text was amended at line 118 (Study design)

We hope that these clarifications address the concerns raised and provide a clearer understanding of the study design and scope.

• How was sampling conducted? Is it purposive, convenient, or what?

R= Sampling was conducted using a non-probabilistic convenience approach. Postpartum WLWH receiving ART treatment as part of the infection care, over 18 years old, with prenatal care at the INPer, were invited to participate during the visits of baby´s follow-up. The text was amended at lines 137-139. Women enrolment section.

• The inclusion criteria are not clear? Why was ART treatment included? Have you applied any measures to monitor and control the ART intervention?

Answer. Concerning the inclusion criteria, it is clarified that eligible participants were postpartum women living with HIV, aged over 18 years, who were receiving ART as a part of their infection care and had received prenatal care at the National Institute of Perinatology (INPer) and just came back only for their baby's follow-up until obtaining discharge, upon confirmation of no HIV vertical transmission (18 months). The selection of these criteria was driven by two overarching objectives. Firstly, it was deemed essential to ensure a homogeneous study population, thereby facilitating the analysis and enhancing the interpretability of the results. Secondly, the criteria were chosen to reflect the clinical reality of women routinely managed at INPer, ensuring the relevance and applicability of the findings to real-world scenarios. You can find in the section that corresponds to it in the Materials and Methods.

About the issue of the rationale behind the incorporation of ART, it is crucial to underscore that this was not an intervention initiated by the study; rather, it was the standard treatment that all participants were already receiving as part of their routine medical care. As a result, ART was considered a significant clinical condition and relevant contextual factor in the analysis, given its recognised influence on health outcomes in women living with HIV.

Finally, regarding the question of whether any measures were implemented to supervise or control ART, it is important to note that no additional monitoring or intervention was performed within the framework of this study. The present study concentrated exclusively on the evaluation of quality of life and its possible influence from sociodemographic and clinical variables, without implementing any modifications or interventions in the participants' treatment.

• Please justify why the sampling was performed during pregnancy, and data collection was performed after delivery.

Answer. We appreciate your careful observation. The potential confusion may have arisen because woman living with HIV are identified during pregnancy at our institute or are transferred from other health institutions for prenatal care following an HIV diagnosis. However, enrolment was carried out exclusively in the postpartum period, when these women attended their infants' routine follow-up visits at INPer. Our research focused specifically on postpartum women, and therefore the invitation to participate was extended at that stage.

We have made the necessary corrections to the manuscript. These ensure that the enrolment process is described accurately and consistently with the study's focus on postpartum women. See Materials and Methods section.

• How was the ART intervention tracked?

Answer. We would like to clarify that ART was not an intervention assigned, introduced, or supervised by the study or the research group, but rather the standard treatment routinely provided to all women living with HIV at the National Institute of Perinatology (INPer) or the ART that patients referred from other medical institutions were already receiving. Accordingly, no specific measures were implemented within the framework of this research to monitor or control the administration of ART.

ART was therefore considered a clinical factor relevant to the analysis, but it was not modified, supervised, or controlled by the investigators, and this study is purely observational.

• Have you maintained follow-up of the women from recruitment to the actual data collection? if yes, how was the follow-up maintained?

Answer. We would like to clarify that no follow-up was conducted from recruitment to data collection. The study was designed as a cross-sectional assessment. Therefore, the quality of life instrumen

---

## [Editor Report · Decision Letter 1]

24 Apr 2026

Quality of life in the postpartum period of Mexican women living with HIV: The role of clinical and sociodemographic factors

PONE-D-25-40652R1

Dear Dr. Jessica Hernández-Pineda,

We’re pleased to inform you that your manuscript has been judged scientifically suitable for publication and will be formally accepted for publication once it meets all outstanding technical requirements.

Kind regards,

Marwan Salih Al-Nimer, MD, PhD

Academic Editor

PLOS One
---

## [Editor Report · Acceptance letter]

PONE-D-25-40652R1

PLOS One

Dear Dr. Hernández-Pineda,

I'm pleased to inform you that your manuscript has been deemed suitable for publication in PLOS One. Congratulations! Your manuscript is now being handed over to our production team.

Kind regards,

on behalf of

Professor Marwan Salih Al-Nimer

Academic Editor

PLOS One